



# High Bandwidth Measurements of Auroral Langmuir Waves with Multiple Antennas

Chrystal Moser[1], James LaBelle[1], and Iver H. Cairns[2]

[1]Department of Physics and Astronomy, Dartmouth College, Hanover, NH, USA
[2]School of Physics, University of Sydney, Sydney, AU

**Correspondence:** Chrystal Moser (Chrystal.Moser.GR@dartmouth.edu)

**Abstract.** The High-Bandwidth Auroral Rocket (HIBAR) was launched from Poker Flat, Alaska on January 28, 2003 at 07:50 UT towards an apogee of 382 km in the night-side aurora. The flight was unique in having three high-frequency (HF) receivers using multiple antennas parallel and perpendicular to the ambient magnetic field, as well as very low frequency (VLF) receivers using antennas perpendicular to the magnetic field. These receivers observed five short-lived Langmuir wave bursts lasting from 0.1–0.2 s, consisting of a thin plasma line with frequencies in the range of 2470–2610 kHz that had an associated diffuse feature occurring 5–10 kHz above the plasma line. Both of these waves occurred slightly above the local plasma frequency with amplitudes between 1–100 $\mu$V/m. The ratio of the parallel to perpendicular components of the plasma line and diffuse feature were used to determine the angle of propagation of these waves with respect to the background magnetic field. These angles were found to be comparable to the theoretical Z-infinity angle that these waves would resonate at. The VLF receiver detected auroral hiss throughout the flight at 5–10 kHz, a frequency matching the difference between the plasma line and the diffuse feature. A dispersion solver, partially informed with measured electron distributions, and associated frequency- and wavevector-matching conditions were employed to determine if the diffuse features could be generated by a nonlinear wave-wave interaction of the plasma line with the lower frequency auroral hiss waves/lower-hybrid waves. The results show that this interpretation is plausible.

## 1 Introduction

Plasma waves generated at or near the local plasma frequency have been observed in the auroral ionosphere by satellites and rockets ever since there have been instruments capable of measuring them [review by Akbari et al. 2020]. These wave amplitudes can range from a few mV/m [McFadden et al., 1986] to greater than 1 V/m [Kintner et al., 1995] and have been observed in both under- ($f_{pe} < f_{ce}$) and over-dense ($f_{pe} > f_{ce}$) plasmas, where $f_{pe}$ is the electron plasma frequency and $f_{ce}$ is the electron cyclotron frequency [Beghin et al., 1989; McAdams et al. 1999]. Simultaneous observations of electron distribution functions and plasma waves have been reported by McFadden et al. [1987], Ergun et al. [1991a] and Beghin et al. [1989], who also showed that frequency structures within the waves occur often in the auroral ionosphere, with an 80% occurrence rate on the dayside and 60% on the nightside. More recent observations of Langmuir waves by the TRICE-1 (Twin Rockets to





Investigate Cusp Electrodynamics) sounding rocket were reported by LaBelle et al. [2010], with modulations as low as 1 kHz
and up to tens of kHz in an underdense plasma.

McAdams & LaBelle [1999] and Samara & LaBelle [2006] observed structured spectral peaks above the plasma frequency
in High-Frequency (HF) spectrograms. The former dubbed these bursts "chirps", with amplitudes up to 1 mV/m relatively
close to $f_{pe}$, and with similar amplitude diffuse waves occurring above the chirp signal. The latter reported several similar
observations made by the SIERRA (Sounding of the Ion Energization Region: Resolving Ambiguities), PHAZE II (Physics of
Auroral Zone Electrons), and RACE (Rocket Auroral Correlator Experiment) sounding rockets, all of which were in an over-
dense plasma. These were investigated theoretically by McAdams et al. [2000] who interpreted them as linear eigenmodes in
pre-existing density structures. Similar Langmuir eigenmodes have subsequently been observed in the solar wind (Malaspina
et al. 2008; Ergun et al. 2008).

Evidence for nonlinear processes has been reported, as recently reviewed by Akbari et al. [2020]. Stasiewicz et al. [1996],
using Freja satellite data, observed evidence of both parametric decay of a Langmuir wave into a lower hybrid ($LH$) and an
oblique wave ($L'$), via the process $L \rightarrow L' + LH$, and scattering off an existing LH wave ($e.g., L + LH \rightarrow L'$), confirmed by
Lizunov et al. [2001] and Khotyaintsev et al. [2001]. A model based on scattering of the plasma wave with an electrostatic
whistler/lower hybrid wave is put forth as a plausible explanation for the modulations observed by Freja and SCIFER (Sounding
of the Cusp Ion Fountain Energization Region) [Bonnell et al., 1997]. Cairns and Layden [2018] reviewed the decay process
of generalized Langmuir waves into backscattered Langmuir waves and either ion acoustic waves or ion cyclotron waves,
and showed, in a strongly magnetized plasma ($f_{pe} < f_{ce}$), the backscattered Langmuir wavenumber is greater than the initial
Langmuir wavenumber, $k_{L'} > k_L$.

McFadden et al. [1986] measured both parallel and perpendicular components of the electric field, observing Langmuir
waves with larger parallel components such that $k_{||} > k_{\perp}$, that were coincident with unstable parallel electron distributions.
Colpitts and LaBelle [2008] performed a Monte Carlo simulation of the Langmuir and Z-mode waves and showed their electric
fields are preferentially parallel, becoming more perpendicular as the frequencies increased towards the UH frequency as
expected. Dombrowski et al. [2012] used the unique 3-D data set from TRICE-1 to determine the intensity of the electric field
for Langmuir waves and shows their parallel components are more than two times larger than their perpendicular components.

The High-Bandwidth Auroral Rocket (HiBAR) was one in a series of sounding rockets equipped with the telemetry capable
of measuring high frequencies waves in detail. Uniquely, it achieved these measurements in both the parallel and perpendicular
direction with respect to the background magnetic field. Its goal was to measure waves generated by intense beams of electron
precipitating down the magnetic field at high latitudes in the F-region of the ionosphere, where $f_{pe} < f_{ce}$. Previously, Samara et
al. [2004] analyzed UH waves from HIBAR at the condition $f_{UH} = 2f_{ce}$, where $f_{UH}$ is the upper hybrid (UH) frequency, the
source of auroral roar emissions seen at ground level [review by LaBelle and Treumann 2002]. This report presents observations
by the HIBAR mission of Langmuir wave bursts near $f_{pe}$, with a region of diffuse waves occurring at a frequencies 5-15kHz
above the plasma bursts, as well as low frequency whistler mode hiss occurring between 5–15 kHz. The wave events are
observed in the overdense regime. Using a wave dispersion solver to determine the normal modes of the waves and the growth



rates for the normal modes, we will show these waves could plausibly be generated by a wave-wave interaction of the Langmuir wave with low frequency waves in the Lower-Hybrid mode.

## 2  Data Presentation

HIBAR was launched from Poker Flat, Alaska, on January 28, 2003, at 07:50 UT into active pre-midnight aurora, reaching an apogee of 382 km. Its payload included a Langmuir probe, particle detectors, and DC, VLF and HF electric field receivers. HIBAR was one in a series of rockets with a high telemetry rate to measure waves with frequencies up to 5 MHz, allowing observations of detailed structure of high frequency waves in the lower ionosphere, such as Langmuir and Upper-Hybrid (UH) waves. The rocket's spin axis was aligned to within 5 degrees of the background magnetic field, with a spin rate of 0.95 Hz. For wave measurements, the rocket included two radial booms oriented perpendicular to one another and three axial booms, one along the axis of the rocket protruding from the front deck, and two mounted on the ends of the radial booms (see Figure 1).

The unique feature of HIBAR was the large number of HF telemetry links. Among these, two were dedicated to measurements of components of HF wave electric fields up to 5 MHz: the perpendicular electric field used probes $x_1$ and $x_3$, located 2.5 m apart oriented perpendicular to the rocket axis, and the parallel electric field used probes $x_1$ and $x_2$, located 0.28 m apart and oriented along the rocket axis. Voltage differences between these probe pairs, amplified and filtered, modulated dedicated transmissions from rocket to ground station. An automatic gain control (AGC) was used to optimize dynamic range. The AGC level was transmitted as a separate PCM link and combined with the HF signal in post analysis. Four electrostatic analyzers (ESA) measured ion and electron energies from 70 eV to 19 keV at 8 pitch angles from 0°–180°, sweeping through the energy steps every 45 milliseconds.

Figure 2a–b shows HF spectrograms from both perpendicular and parallel antennas covering 07:54:13–07:54:33 UT (253–273 s) flight time and the altitude range to ∼364–374 km, one of the intervals when Langmuir waves were observed. Figure 2c–d show data for a slightly later interval, 07:55:49–07:56:09 UT (349–369 s), corresponding to 377–370 km altitude, which also contains Langmuir waves. As usual, plasma noise is enhanced in the band between $f_{pe}$ and $f_{UH}$, so that the local plasma frequency can be seen as lower cutoffs in both the spectrograms between 2400 and 2700 kHz, and the upper-hybrid frequency can be seen as an upper cutoff in the perpendicular spectrograms between 2800 and 3000 kHz. During these two time intervals, HIBAR encountered seven short-lived wave bursts near $f_{pe}$ that last from ∼ 0.1 − 0.2 s, five of which had a diffuse band occurring 5-10kHz above a narrow plasma wave line (see Figure 4) and well below the upper hybrid band above 2800 kHz. These five events are labeled in Figure 2 by their respective times (in seconds after launch), occurring at 07:54:20, 07:54:22, 07:54:32 in Figure 2a–b and 07:55:51 and 07:55:59 UT in Figure 2c–d. For the entirety of both intervals in Figure 2, HIBAR is in overdense plasma ($f_{pe} > f_{ce}$).

Figure 3 shows the Very-Low Frequency (VLF) data in a frequency-time spectrogram for the interval where the Langmuir wave bursts are seen, between 07:54:10–07:56:10 UT (250–370 s) and ∼360–380 km. There is a broadband enhancement of the whistler mode waves between 4–15 kHz, with a small band of slightly more enhanced waves at approximately 5 kHz,





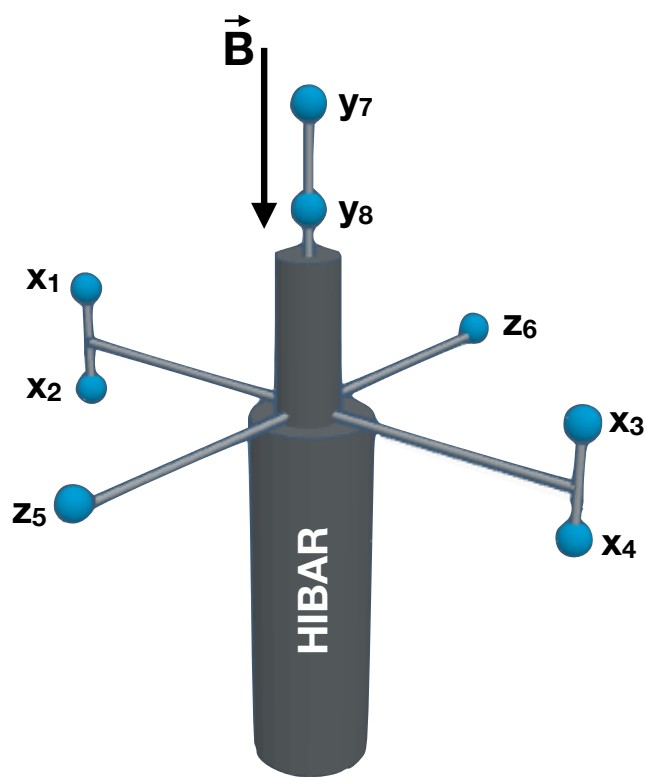

**Figure 1.** Diagram of the HIBAR rocket showing approximate antenna orientations with respect to the background magnetic field. (note: the labeling of the probes has no connection to Cartesian coordinates.)

believed to be near the LH frequency because it acts as a cutoff to the whistler mode. These waves were measured with a separate perpendicular antenna, oriented 90° to the antennas used to measured the HF waves, using probes z5 and z6 in Figure 1.

Figure 4 shows enhanced spectrograms of five selected Langmuir wave events observed by both the parallel and perpendic-
ular HF antenna labeled 260s, 262s, 271s, 351s, and 359s in Figure 2. These events include a thin, intense plasma line just above the plasma frequency cutoff and a less intense band of waves above the plasma line, referred to as the diffuse feature. Other plasma line events occurred during HIBAR; however, these did not include the diffuse waves, and therefore were not considered in this study. Obtaining absolute units for the electric fields of these features requires combining the AGC voltage data with the raw HF waveform data. These values where then divided by the length of the respective booms to obtain electric
fields in V/m, under the assumption, discussed below, that the wavelength is longer than the probe separation.

Black boxes in each spectrogram in Figure 4 outline time and frequency intervals used to calculated average intensities of the plasma line and diffuse features of each event. Figure 5 shows details of this calculation for a selected event, shown in Figure





**Figure 2.** 2000-3200 kHz spectrograms of perpendicular (upper panels a & c) and parallel (lower panels b & d) HF electric fields for two time intervals during the HIBAR rocket flight: 07:54:18–07:54:33 UT and 07:55:49–07:36:04 UT, showing the plasma frequency cutoff as a lower bound in the perpendicular and parallel spectrograms, and the upper-hybrid frequency cutoff as an upper bound in the perpendicular spectrograms. Red circles indicate five Langmuir wave bursts used for detailed study.

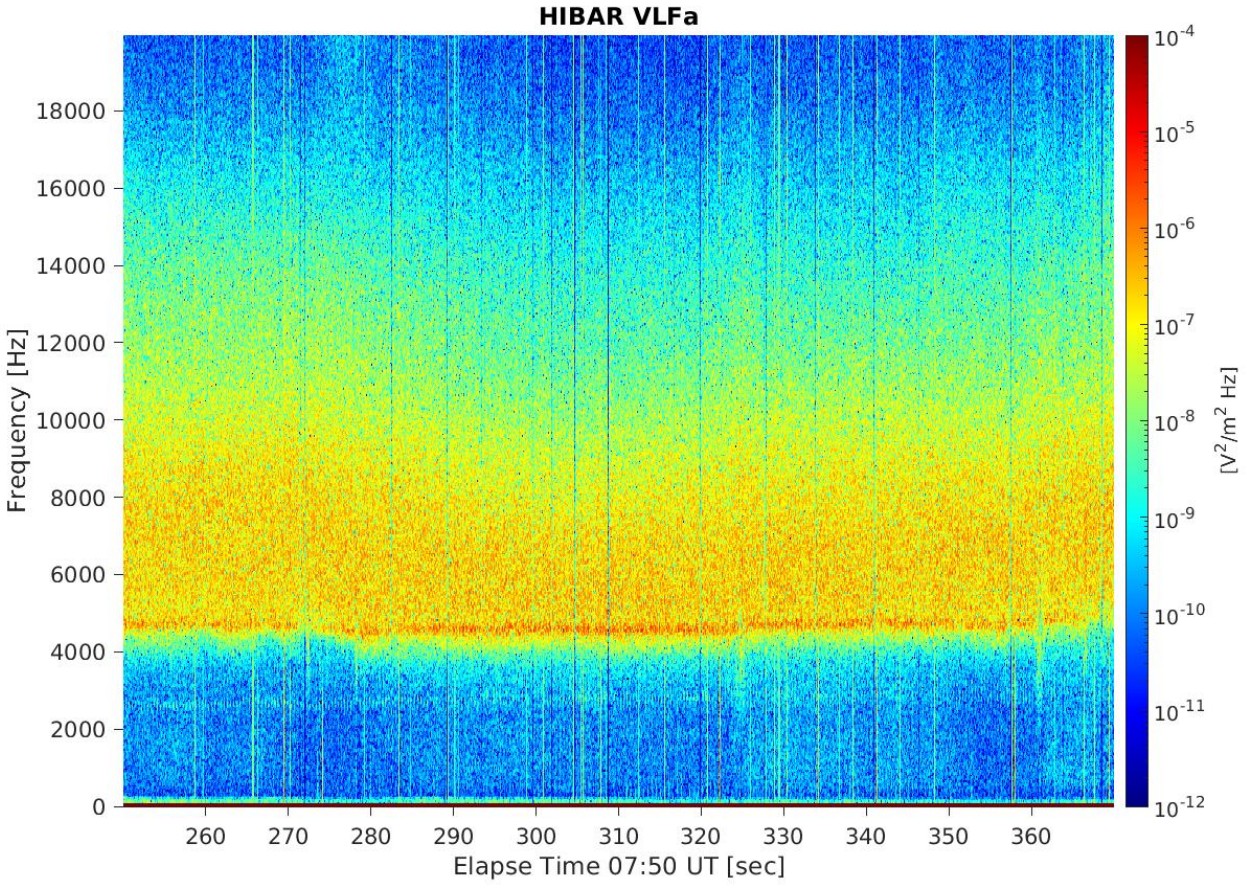

**Figure 3.** Frequency-Power spectrogram of the HIBAR VLF wave data from 0-20 kHz and 07:54:10–07:56:10 UT (250–370 s) showing the broadband diffuse whistler mode waves, and a slightly enhanced power band at ∼5 kHz corresponding to probable LH waves.

4a as occurring at 259.9–260.0 s. Separately for both the parallel and perpendicular spectra, the background power spectral density level was determined for each event by computing the average spectral density over a slightly higher frequency range, as indicated by the upper black box spanning 2640–2660 kHz in Figure 5a–b. The background interval was selected separately and was slightly different for each of the other four events shown in Figure 4. For each event, a spectrum was produced by subtracting this average background power spectral density from each spectrum. Figure 5c shows example spectra after this subtraction, for both perpendicular (blue trace) and parallel (red trace) for the time indicated by a red vertical line in Figure 5a–b. This was done because the background noise, either from the instrument or from the environment, was significantly different between the two antennas, and would have effected the ratio of the electric fields. It was removed for a more accurate estimate of the parallel to perpendicular values.



**Figure 4.** Enhanced plots of the five Langmuir bursts indicated in Figure 2, presented in time order, each comprised of a narrow band plasma line and a broadband diffuse feature with ∼5–15 kHz higher frequency. The top panels in each plot are from the perpendicular antenna, the middle panels are from the parallel antenna, and the bottom panels are the parallel to perpendicular ratios of the amplitudes of the plasma peaks (red) and the diffuse feature (blue).





The average intensity of each feature for each antenna is determined by integrating the appropriate spectrum over the frequency range of the feature, bounded by the vertical dashed line in Figure 5c, corresponding to the black boxes in Figure 4 and Figure 5a–b. In the case of the selected event shown in Figure 5, the intensity is 7.8E-9 $V^2/m^2$ (4.6E-10 $V^2/m^2$) for the plasma line with the parallel (perpendicular) antenna, and 3.3E-10 $V^2/m^2$ (5.2E-11 $V^2/m^2$) for the diffuse feature with the parallel (perpendicular) antenna. These numbers combine to imply that $E_{||}/E_\perp$ is $2.3 \pm 1.2$ for the plasma wave and $2.0 \pm 0.4$ for the diffuse wave when averaged over the whole interval of the event shown in Figure 5, with the standard deviation specified.

Bottom panels of each section of Figure 4 display $E_{||}/E_\perp$ ratios for both the plasma line (red points) and diffuse feature (blue points) as a function of time through the five selected events. For the plasma line, the variation in this ratio is noteworthy: it seems to toggle between a fairly high ratio, around five, and a low ratio near unity. There is no obvious feature in the spectrograms mirroring these changes in the polarization state, leading us to investigate the theoretical or instrumental reason for this unexpected result (discussed below). Because of this non-stationarity of the polarization, $E_{||}/E_\perp$ ratios averaged over the entire event may be misleading. Table 1 summarizes the polarization measurements of each event shown in Figure 3. The table has seven rows because two of the events, at 351 and 359 s, have been split into two events, as indicated by the black boxes in Figures 4d–e, because they each have a gap in the plasma line suggesting they may be two events in close proximity. Table 1 tabulates both the average $E_{||}/E_\perp$ ratio, which may be misleading as discussed above, the maximum $E_{||}/E_\perp$ ratio defined as the average of the highest three measured ratios, and the minimum $E_{||}/E_\perp$ ratio defined as the average of the lowest three measured ratios for consistency. Uncertainty estimations are based on standard deviations associated with the averages taken in obtaining each $E_{||}/E_\perp$ value.

## 3 Discussion

The mean $E_{||}/E_\perp$ ratios in Table 1 for the plasma line range from 1.8 to 5.4 and average 2.9, in approximate agreement with previous measurements which had generally lower time resolution. For example, McFadden et al. [1986] reported ratios ranging from 3-10. As noted by McFadden et al. [1986], wavelength as well as polarization can affect the measured ratio $E_{||}/E_\perp$. In the case of HIBAR, electrons measured with the ESA had relatively high energy, in the range 10-20 keV. For a plasma frequency of ~2600 kHz, this implies Langmuir waves with parallel wavelengths of ~23–32 m would resonate with the electron distribution measured by HIBAR. Assuming that the standard electron beam Langmuir wave instability for electrons with these energies gives rise to the plasma line implies that the wavelength should exceed the probe separations which were of order 0.3 m for the parallel measurement. The perpendicular measurement used longer boom separation, 3.0 m, but the measured $E_{||}/E_\perp$ ratio suggests that measurement is also in the long-wavelength regime. This means that the wave polarization should be the dominant effect determining the $E_{||}/E_\perp$ ratio for the plasma line.

McFadden et al. [1986] also point out that the perpendicular component of the wave may be underestimated in the measurement by a factor $\cos\phi$, where $\phi$ is the angle between the perpendicular electric field boom and the instantaneous perpendicular wavevector, assuming that the wave has a distinct perpendicular wavevector rather than being distributed over a range of wavevectors during the time of measurement. In the latter case, the perpendicular electric field will be underestimated by a



**Figure 5.** (a) Perpendicular and (b) parallel spectrograms for the Langmuir bursts labeled 260s in Figure 2 and shown in Figure 4a. Black boxes indicate the frequency-time ranges used to define the plasma line, diffuse feature, and background level. (c) Selected spectrum with background noise subtracted, occurring at the time highlighted as a red vertical line in panels (a) and (b), showing the power spectral density of the parallel waves (blue) versus the perpendicular waves (red).





**Table 1.** Mean ratios of $E_{||}/E_\perp$, maximum $E_{||}/E_\perp$ defined as the mean of the three largest ratios for each event, and minimum $E_{||}/E_\perp$ defined as the mean of the three smallest ratios for each event with their standard deviations for both the plasma line ($f_p$) and the diffuse feature ($f_{diff}$), for Langmuir bursts defined in Figures 2–4. Event's 351 and 359 were split into 2 separate events because of the gap in the plasma line in the middle of the event interval.

| Event Time [s] | Mean ratio $E_{||}/E_\perp$ | | Max Ratio $E_{||}/E_\perp$ | | Min Ratio $E_{||}/E_\perp$ | |
| --- | --- | --- | --- | --- | --- | --- |
| | $f_p$ | $f_{diff}$ | $f_p$ | $f_{diff}$ | $f_p$ | $f_{diff}$ |
| 260 | 2.25±1.23 | 1.97±0.44 | 5.01±0.40 | 2.84±0.37 | 0.86±0.19 | 1.35±0.05 |
| 262 | 2.15±1.39 | 1.83±0.43 | 4.49±1.90 | 2.63±0.19 | 1.01±0.13 | 1.36±0.16 |
| 271 | 3.97±2.17 | 2.46±0.39 | 7.38±0.20 | 3.01±0.24 | 1.63±0.37 | 1.92±0.15 |
| 351–1 | 2.45±1.65 | 1.84±0.40 | 5.71±1.10 | 2.50±0.25 | 0.95±0.12 | 1.32±0.07 |
| 351–2 | 1.84±1.96 | 1.71±0.38 | 6.38±2.26 | 2.35±0.14 | 0.41±0.09 | 1.07±0.07 |
| 359–1 | 5.41±1.91 | 2.78±0.38 | 7.73±0.38 | 3.28±0.21 | 2.50±0.84 | 2.23±0.17 |
| 359–2 | 2.02±0.84 | 2.38±0.65 | 3.50±0.34 | 3.32±0.33 | 1.22±0.21 | 1.51±0.18 |





**Table 2.** The resulting angles $\theta$ from Equation (3) for the mean and maximum ratios defined in Table 1 for both the plasma line ($\theta_p$) and diffuse feature ($\theta_{\text{diff}}$).

|  | $E_{||}/E_{\perp}$ | Mean ratio | $E_{||}/E_{\perp}$ | Max Ratio |
| --- | --- | --- | --- | --- |
| Event Time [s] | $\theta_p$ | $\theta_{\text{diff}}$ | $\theta_p$ | $\theta_{\text{diff}}$ |
| 260 | 24° | 27° | 11° | 19° |
| 262 | 25° | 28° | 13° | 21° |
| 271 | 14° | 22° | 8° | 18° |
| 351–1 | 22° | 29° | 10° | 22° |
| 351–2 | 28° | 30° | 9° | 23° |
| 359–1 | 10° | 20° | 7° | 17° |
| 359–2 | 26° | 20° | 16° | 17° |

smaller factor. These considerations raise the question of whether the observed bimodal distributions of $E_{||}/E_{\perp}$, seemingly toggling between high values $\geq 5$ and low values near unity, result from variations in the angle between the perpendicular boom and the wave vector projected into the plane perpendicular to $\boldsymbol{B}$, rather than variations in the fundamental polarization of the waves. In principle, it is impossible to distinguish these two possibilities since both types of time variation of the wave vector could equally well produce the observed $E_{||}/E_{\perp}$ ratios. It is possible to infer, however, that if the angle between the

perpendicular boom and the wave vector projected on the plane perpendicular to $\boldsymbol{B}$ is stationary, the mere rotation of the booms cannot explain the observed variations in $E_{||}/E_{\perp}$ (since the observed variations do not appear to repeat at the spin period).

An attempt to determine the angle of the perpendicular wavevector to the antennas orientation results in poor fits to the observed time series of $E_{||}/E_{\perp}$ (not shown), as the observed data have zero correlation or, in some cases, the exact opposite correlation, to the expected trend based on the fit equations. The time variations in the measured $E_{||}/E_{\perp}$ suggest that either

some aspect of the polarization, the $E_{||}/E_{\perp}$ ratio itself, or the angle of the $E_{\perp}$ vector changes on sub-second timescales, giving rise to variations in the observed value of $E_{||}/E_{\perp}$, or the waves are distributed over some peculiar range of angles such that the rocket spin produces this effect through variation of the angle between the boom and the projection of the electric field vector into the plane perpendicular to $\boldsymbol{B}$. Either way, one may safely infer that $k_{||}$ exceeds $k_{\perp}$ for these waves, as expected for Langmuir waves close to the plasma frequency.

It is worth noting, however, that Langmuir waves driven in the relatively unmagnetized solar wind by electron beams with energies of order 100 keV and above can naturally have $E_{\perp}/E_{||} > 1$ [Graham and Cairns, 2013a; Malaspina and Ergun, 2008]. Theoretically, this situation involves wave growth driven by the electron beam on or at least near the z-mode portion of the generalized Langmuir mode, corresponding to frequencies very near and below $f_{pe}$ [Willes and Cairns, 2000]. The relevant condition on the wavenumbers is





$$k_w^* \lambda_D = \frac{V_e}{c} \left[ cos^2\theta + \frac{\omega_{pe}}{\omega_{ce}} \right]^{1/2} \text{ or} \tag{1}$$

$$k^* = \frac{\omega_{pe}}{c} \left[ cos^2\theta + \frac{\omega_{pe}}{\omega_{ce}} \right]^{1/2} \tag{2}$$

where $k^*$ is the wavenumber, $\omega_{pe}$ is the electron plasma frequency, $c$ is the speed of light, $\theta$ is the angle of the wavevector with respect to the background magnetic field, and $\omega_{ce}$ is the electron cyclotron frequency.

In the HiBAR situation, where $\omega_{pe}/\omega_{ce} \approx 2$, this requires wavenumbers on the order of 0.1 m$^{-1}$. Ignoring semi-relativistic
and magnetization effects, the corresponding speeds are $v = \omega_{pe}/k^* \approx 0.5c$. The corresponding energies are $\sim$70 keV, between the energies of $\sim$10 - 100 keV considered standard for the auroral ionosphere, but beyond the energy range that the electrostatic analyzer could measure. Accordingly at this time, we seek an explanation in terms of slower electron beams.

### 3.1 Electric Field Component Ratios

Theory also suggests that as waves increase in frequency away from the local plasma frequency, they should become more
perpendicular, decreasing the ratio of parallel to perpendicular electric field (see Figure 6). This prediction is confirmed in this study (see Tables 1 and 2), where the ratios $E_{||}/E_\perp$ of the plasma lines exceed those of the diffuse feature that occurs at higher frequencies. This is true for the total average over each event interval ($E_{||} \approx (2 \text{ to } 5)E_\perp$ for the plasma line and $E_{||} \approx 2E_\perp$ for the diffuse feature), and for the average max ratio between the two waves ($E_{||} \approx (4 \text{ to } 8)E_\perp$ for the plasma line and $E_{||} \approx 3E_\perp$ for the diffuse feature). In the extreme case, waves near $f_{UH}$ reported by Samara et al. [2004] have very small $E_{||}/E_\perp$ ratios
with an average of 0.05 (see Figure 2 of Samara et al., 2004).

From the ratios in Table 1 the angle of wave propagation can be calculated using simple geometry by assuming the electric field amplitude ratio is proportional to the wavenumber ratio ($E_{||}/E_\perp = k_{||}/k_\perp$), as expected for electrostatic waves, where the angle with respect to the magnetic field, $\theta$, is given by

$$\theta = 90° - \tan^{-1}\left(\frac{E_{||}}{E_\perp}\right). \tag{3}$$

Table 2 shows calculations of these angles for both the total average ratio and the max average ratio, and for both the plasma line and the diffuse feature.

The unique capability of the HIBAR mission to measure both the parallel and perpendicular components of the electric field means the propagation angles of waves with respect the background magnetic field can be compared to the expected values from plasma theory. Because these waves occur slightly above the plasma frequency cutoff in the overdense plasma ($f_{pe} > f_{ce}$),
they fall into the Z-mode region (see Figure 6 adapted from Benson et al. 2006). In this region, for waves with phase velocities less than c, the waves can experience resonance referred to as the upper oblique resonance given by Benson et al. (2006)

$$f_{ZI} = \frac{1}{\sqrt{2}} \left[ f_{UH}^2 + (f_{UH}^4 - 4f_{ce}^2 f_{pe}^2 cos^2\theta)^{\frac{1}{2}} \right]^{\frac{1}{2}}. \tag{4}$$

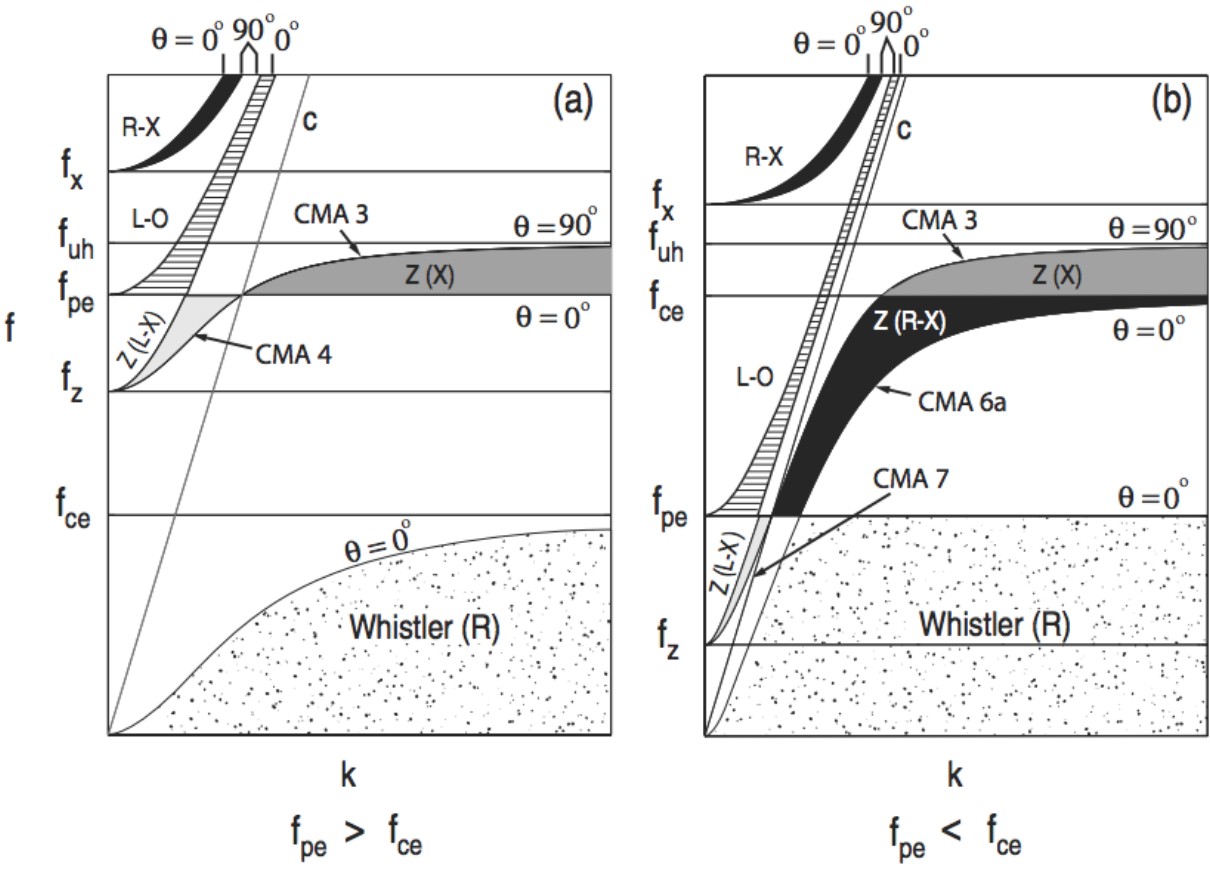

**Figure 6.** Dispersion relations for the different wave modes for an overdense ($f_{pe} > f_{ce}$) and underdense ($f_{pe} < f_{ce}$) plasma, adapted from Benson et al. [2006]. The Z-mode cutoff above the plasma frequency for an overdense plasma increases from 0 to $\frac{\pi}{2}$.

The frequency that waves can resonate in this region, Z-infinity $f_{ZI}$, depends on the local electron plasma frequency $f_{pe}$, the electron cyclotron frequency $f_{ce} \approx 1350$ kHz, and the angle that the wave propagates at with respect to the background magnetic field, $\theta$. In the limit $\theta \to \frac{\pi}{2}$, $f_{ZI} = f_{UH}$, and in the limit $\theta \to 0$, $f_{ZI} = \max[f_{pe}, f_{ce}]$. Table 3 lists the frequencies for the plasma cutoff ($f_{pe}$), the plasma line ($f_p$, assumed to be $f_{ZI}$), and the range of the diffuse feature ($f_{\text{diff}}$) for each wave burst, labeled by when they occurred in seconds post launch, along with the calculated oblique angle of the Z-infinity resonance. The angles calculated from equation (4) agree fairly well with the angles determined from the electric field ratios in equation (3). These angles agree better with the angles calculated from the average of the max power ratios than the average over all power ratios for each event for both the plasma line and diffuse feature, consistent with the non-stationary aspect of these waves. This suggests these waves are resonating at the Z-infinity resonance angle.



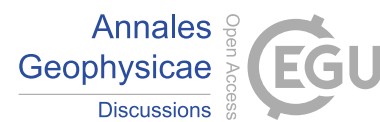

**Table 3.** Plasma frequency cutoff ($f_{pe}$), plasma line frequency ($f_p$), diffuse feature frequency range ($f_{\text{diff}}$), and resonant z-mode oblique angles, $\theta_p$ and $\theta_{\text{diff}}$, calculated from equation (4), for Langmuir bursts labeled in Figures 2–4.

| Event Time [s] | Plasma Cutoff $f_{pe}$ [kHz] | $f_p$ [kHz] | $f_{\text{diff}}$ [kHz] | $\theta_p$ [deg] | $\theta_{\text{diff}}$ [deg] |
|---|---|---|---|---|---|
| 260 | 2586 | 2607 | 2615-2625 | 12 | 14-17 |
| 262 | 2525 | 2540 | 2545-2556 | 10 | 12-15 |
| 271 | 2460 | 2471 | 2475-2488 | 4 | 7-11 |
| 351 | 2575 | 2580 | 2586-2600 | 8 | 11-15 |
| 359 | 2600 | 2606 | 2611-2623 | 9 | 11-14 |

**Table 4.** Parameters used for computing dispersion surfaces in WHAMP associated with Langmuir bursts labeled in Figures 2–4.

| Event Time [s] | B [nT] | n [cm$^{-3}$] | $T_{||}$ [eV] |
|---|---|---|---|
| 260 | 48402 | 82953 | 0.2 |
| 262 | 48345 | 79337 | 0.2 |
| 271 | 48202 | 75128 | 0.2 |
| 351 | 48074 | 81294 | 0.2 |
| 359 | 48380 | 83854 | 0.2 |

## 3.2 Non-Linear 3-Wave Interaction

The plasma lines and corresponding diffuse features last for identical time intervals. This raises the possibility that the diffuse features are generated by wave-wave interactions of the plasma lines with lower frequency waves. HIBAR was equipped with a very-low frequency (VLF) receiver that measured waves from 0– 20 kHz, which showed a consistent whistler mode hiss for the times when the HF waves are observed (e.g. Figure 3). The whistler hiss ranges from 5-15 kHz and has wave electric fields on the order of tens of mV/m. The broad range of whistler waves surrounding the rocket could interact with the plasma line to generate the broad range that the diffuse wave exhibits.

To test the plausibility of the wave-wave interaction hypothesis, a dispersion solver, Wave in Homogeneous Anisotropic Multicomponent Plasma (WHAMP, Rönnmark 1982), was employed to calculate surfaces corresponding to the normal modes in the plasma that might participate in the wave-wave interaction: the Langmuir-Upper Hybrid (UH) and the Whistler-Lower Hybrid (LH) surfaces. WHAMP requires user defined input parameters for the plasma environment, including the magnetic field strength, number of particle species and their respective densities and temperatures. Table 4 lists the parameter values used for modeling each HIBAR event. The two species used were electrons and oxygen ions, which are the dominant ions at low altitudes, and each were represented by a basic Maxwellian distribution. The densities were determined from the plasma frequency cutoff, and the magnetic field from the magnetometer on board the rocket. Temperatures were taken to be 0.2 eV, typical of auroral F-region, and assumed to be isotropic.





Figure 7 shows the WHAMP surfaces for each of the 5 events, where the x and y axes are the perpendicular and parallel wavenumbers normalized to the electron gyroradius, and the z axis is the wave frequency normalized to the electron gyrofre-
220 quency. For the Langmuir-UH surface, in the parallel wavenumber limit the frequency equals the electron plasma frequency and in the perpendicular limit the frequency equals the upper-hybrid frequency. For small wavenumbers ($\rho_{||}k_{\perp} < 10^{-2}$) this surface corresponds to the Z-mode (cf Willes and Cairns, 2000). On the Whistler-LH surface, in the large parallel wavenumber limit ($k_{||} \gg k_{\perp}$) the frequencies approach the electron cyclotron frequency. The LH surface is found at near perpendicular propagation ($k_{\perp} \gg k_{||}$). At oblique angles near parallel to $\boldsymbol{B}$ ($k_{||} > k_{\perp}$), the surface corresponds to the whistler mode.

For each Langmuir-UH surface in Figure 7 the black (white) line represents the values of $k_{||}/k_{\perp}$ inferred from the average of the maximum $E_{||}/E_{\perp}$ ratios listed in Table 1 for the plasma lines (diffuse features). The widths of these lines are determined by the standard deviations of the ratio. The corresponding plasma line and diffuse feature frequencies are plotted as patches of yellow and pink, respectively. For each of the plasma line and diffuse feature, where the line for $k_{||}/k_{\perp}$ intersects the patch for the observed wave frequency is the locus of allowed frequencies and wavevectors on the normal mode surface. The red line
represents where $\rho_{||}k_{||}$ corresponds to 20 keV, the maximum electron energy observed by the electrostatic analyzer during the time of the events, via the relationship $k = \omega\sqrt{m_e/2E}$. If the plasma lines were generated by parallel Landau resonance with these high energy electrons, then where the black plasma line ratio and yellow frequency patch intersect should be close to the condition represented by the red line. This occurs for events labeled 260s, 271s, 351s-1, 351s-2, and 359s-1.

Assuming a nonlinear 3-wave interaction is responsible for the generation of the diffuse feature, the possible third wave
should be connected through the wavevector matching condition, $\boldsymbol{k}_3 = \boldsymbol{k}_{\text{diff}} - \boldsymbol{k}_p$, which results from momentum conservation in the interaction [e.g., Tsytovich, 1970; Melrose, 1980; Cairns, 1987, 1988; Cairns and Layden, 2018; Moser et al., 2021]. The wavenumbers $\boldsymbol{k}_p$ and $\boldsymbol{k}_{\text{diff}}$ are determined by the two intersections of wavenumber ratio (black and white) and frequency matching (pink and yellow) on the Langmuir-UH surface. The dark blue patch on the whistler/LH surface in each panel of Figure 7 represents the range of k-vectors on the whistler/LH surface that satisfies this condition. The three modes must also
obey the frequency matching condition, $\omega_3 = \omega_{\text{diff}} - \omega_p$. Light blue points within the region of possible k-vectors for the third wave represent modes that also satisfy the frequency matching condition. All events have a possible third wave that could interact with the plasma line to generate the diffuse feature. In each case Figure 7 suggest the third wave is well-described as a whistler/LH wave.







**Figure 7.** WHAMP dispersion surfaces for Langmuir bursts labeled in Figures 2–4, with $k_{||}/k_{\perp}$ ratios inferred from the maximum $E_{||}/E_{\perp}$ in Table 1 plotted as black for the plasma line and white for the diffuse feature. The yellow and pink areas indicate where the surface matches the frequency of the plasma line and diffuse feature, respectively. Where these intersect defines the range of possible k-vectors for each wave. Assuming wave-wave interaction, kinematic equations imply a range of k-vectors for the possible third wave plotted in dark blue on the whistler/LH surface, and the matching frequency of the third wave plotted in light blue.



These waves were produced by some form of energy exchange of particles with the plasma environment, and the electron and
245 ion data were examined to determine the source of these waves. Similar to the analysis of growth rates in Moser et al. [2021b],
the electron and ion distribution functions are needed to determine growth rates on the two dispersion surfaces produced by
WHAMP. The measured electron distribution for the time 07:54:19.907 UT is shown in Figure 8a, for event labeled 260s,
with a model of the high energy electron distribution in Figure 8b produced by the WHAMP parameters: temperature, density,
magnetic field strength, drift velocity, and anisotropy. The x-axis represents the parallel velocity, where the positive axis is
250 along the background magnetic field and the negative axis is anti parallel to the magnetic field. The y-axis represents the
velocity perpendicular to the background magnetic field. The high energy electrons, while not the most prominent feature
in the electron distribution, were used to model the distribution because equation (2) suggests these waves are produced by
particles with higher energies. It should be noted that the electron ESA could only measure electrons with energies below 20
keV, which limits the range of electron energies that can be modeled.

Figure 8c shows the whistler/LH mode surface produced in WHAMP with growth rates from the model distribution in Figure
8b for the event labeled 260s. The model distribution has a parallel temperature $T_{||}$ = 50 eV, density $n = 1$ cm$^{-3}$, magnetic
field $\boldsymbol{B} = 48402.0$ nT, a drift velocity $v_D = 5u_{||}$, and an anisotropy ratio of $T_{||}/T_{\perp}$ = 5. There are two areas of growth that
are of interest, at low $k_{\perp}$ and high $k_{\perp}$, where the frequency and wavenumber matching conditions are met. At low $k_{\perp}$ the
growth rate are $\sim 10^{-8}$ Hz, smaller than the growth rates at higher $k_{\perp}$ of $\sim 10^{-6}$ Hz, but both are too low to likely produce
these waves. However, the true unstable distributions may not be captured with the particle instruments, even with proper
energy range and resolution, because unstable distributions rapidly stabilize. So while the growth rates with the observed
distribution are low, they show that growth should occur and could increase to non-linear levels with a more suitable electron
distribution. The areas of larger growth at higher frequencies near $k_{\perp}\rho_{||} = 10^{-2}$ on the whistler mode surface are potentially
generating the whistler modes waves observed in the HF spectra at frequencies between about 50 and 350 kHz. The model
electron distribution in Figure 8 was also used to generate the Langmuir/z mode surface (not shown) and found to produce no
instabilities at frequencies and wavenumbers that correspond to the modes in Figure 7.

**Figure 8.** Measured electron distribution function from HIBAR's electron ESA data at 07:54:19.907 (top left) and model of the high energy beam seen in the measured distribution using a drifting Maxwellian (top right). The bottom panel shows the growth rates on the whistler/LH mode surface produced by the model WHAMP distribution along with the frequency and wavenumber matching conditions for the event labeled 260s in pink. The Langmuir/z mode surface showed no growth on the surface from this distribution.

Other possible sources of free energy are electrons above 20 keV and below 60 eV as well as the ions. Because the high and low energy electrons were not measured, they could not be modeled with WHAMP to find unstable features. As stated above, the instability that would be the source of the observed Langmuir waves may result from higher energy electrons than those that were measured. The ions were measured from 80 eV to 20 keV with a time resolution of 45 ms. In a similar analysis to that described above, the observed ion ring-like distribution at 09:54:19.920 UT was modeled using the WHAMP parameters, and growth rates on the whistler/LH modes were analyzed. The resulting model produced low growth rates on the surface ($< 10^{-7}$





Hz), but at wavevectors and frequencies that do not match those seen in Figure 7. Therefore, the ions are unlikely to be the source of these waves.

Another test of plausibility for a wave-wave interaction is to compare the electric energy density of the different waves to the thermal plasma energy density. The electric energy density, $\frac{1}{2}\epsilon_0 E^2$, for the plasma line is $\sim$1E-21 J/m$^3$ and for the diffuse band is $\sim$1E-23 J/m$^3$, 100 times smaller than the plasma line. The whistler/LH mode waves (likely dominated by whistler mode hiss) has an electric energy density of approximately 1E-16 J/m$^3$. In comparison the plasma's thermal energy density is $nk_BT \approx$ 3E-9 J/m$^3$, where $n \sim$ 8E4 cm$^{-3}$ is the plasma number density and $k_BT = 0.2$ eV is the temperature assumed for

all events. The ratio of the electric to the thermal energy densities is $\sim$1E-12 for the plasma line, 1E-14 for the diffuse band, and 1E-7 for the whistler/LH mode hiss. Because the diffuse feature is much weaker than the plasma line and the whistler/LH mode hiss, it suggests that the diffuse feature is a product of a wave-wave coalescence process ($W + L \rightarrow L'$) between the two others, the plasma line ($L$) and whistler/LH mode hiss ($W$). The whistler/LH mode energy density being much larger than the other two suggests that this is the primary driving wave, and the "plasma line" Langmuir waves are secondary, with the diffuse

band being a product wave.

A more quantitative analysis is to examine the ratio of wave occupation numbers for these waves. The electric energy density is related to the plasmon occupation number through

$$\frac{1}{2}\epsilon_0 E^2 = \int\int\limits_{k_{min}}^{k_{max}} \frac{2\pi k_\perp dk_\perp dk_{||}}{(2\pi)^3} \hbar\omega_i(\mathbf{k})R_i(\mathbf{k})N_i(\mathbf{k}) \tag{5}$$

where $R_i(\mathbf{k})$ is the ratio of the electric to total energy, $N_i(\mathbf{k})$ is the occupation number, and the volume integral is over the

relevant region of wavevector space for a participating set of waves (e.g for the plasma line). The ratios $R_i(\mathbf{k})$, as determined by WHAMP, are approximately $\frac{1}{2}$ for both the plasma line and diffuse feature, and $\frac{1}{50}$ for the whistler mode hiss. For the plasma line combining this value of $R_i(\mathbf{k})$ with the electric energy density observed leads to a total energy density of approximately 2E-21 J/m$^3$. The same procedure leads to total energy densities of 2E-23 J/m$^3$ and 5E-15 J/m$^3$ for the diffuse waves and the VLF whistlers, respectively.

Assuming the occupation numbers are the same for each wave mode, equation (5) can be rearranged and the ratios of occupation numbers determined to be

$$\frac{N_L}{N_W} = \frac{\frac{1}{2}\epsilon_0 E_L^2 \omega_W R_W \left[\int\int k_\perp dk_\perp dk_{||}\right]_W}{\frac{1}{2}\epsilon_0 E_W^2 \omega_L R_L \left[\int\int k_\perp dk_\perp dk_{||}\right]_L} \approx 8\text{E-}10 \frac{\left[\int\int k_\perp dk_\perp dk_{||}\right]_W}{\left[\int\int k_\perp dk_\perp dk_{||}\right]_L}. \tag{6}$$

The difficulty with solving this equation is determining the range of wavevectors that the modes occupy. To get a rough estimate of the ranges, the WHAMP surfaces are examined to determine possible ranges of wavenumbers for the observed

waves and get an idea for the ratio of the occupation numbers. For the plasma line and diffuse feature, the broad range of wavevectors is $\rho_{||}k_{||} =$ 1E-3–1E-2 and $\rho_{||}k_\perp =$ 2E-4–2E-3. For the whistler/LH mode the wavevector range $\rho_{||}k_{||} =$ 1E-4–1E-2 and $\rho_{||}k_\perp =$ 2E-5–1E-4, where $\rho_{||} = 0.03$ m. This covers the square patch of the surface where the different wave modes occur that match the conditions in Figure 7. Choosing these ranges in the wavevector integrals in equation (7) leads approximately to





$$\frac{N_L}{N_W} \approx 8\text{E-}10\frac{2\text{E-}6}{6\text{E-}4} \approx 2\text{E-}11. \tag{7}$$

Following a similar derivation for the time rate of change of the occupation numbers as in Moser et al. [2021], Cairns [1988], and Melrose [1980], among others, we can show that at saturation (when the rates of change of $N_L$ and $N_W$ are zero, ignoring linear growth and damping) the relationship of the whistler/LH mode occupation number to the Langmuir wave occupation numbers for the coalescence process is

$$N_W(N_L - N_{L'}) - N_L N_{L'} \simeq 0 \tag{8}$$

$$N_{L'} \simeq \frac{N_L N_W}{N_L + N_W}. \tag{9}$$

For each plasmon lost from the whistler/LH mode and the plasma line as the coalescence $L + W \rightarrow L'$ proceeds,, the diffuse mode gains one plasmon. From equation (9) the process saturates when

$$N_{Lo'} \approx \min(N_W, N_L). \tag{10}$$

This leads to a very small ratio of the Langmuir mode occupation numbers to the whistler/LH mode, with $N_{L'} \approx N_{Lo'} \approx N_L$

when $N_L \ll N_W$, which we've shown is the case from equation 7 for the observations.

Based on the foregoing observations and theoretical analyses it appears plausible that the diffuse band is formed by the nonlinear coalescence $L + W \rightarrow L'$ of whistler/LH mode waves W near the LH frequency with Langmuir waves L. The presumption is that the L and W waves are produced by distinct linear instabilities, most likely driven by an electron beam and/or by temperature anisotropies.

**4   Conclusions**

The HIBAR rocket was launched into active pre-midnight aurora and observed seven short duration bursts of Langmuir waves above the local plasma frequency at altitudes from 364-377 km. Of these seven events, five consisted of a plasma line at frequencies ranging from 2470–2610 kHz with an associated diffuse feature occurring 5–15 kHz above this line. Independent measurements of both the parallel and perpendicular components of the electric field showed that the plasma lines typically

have $E_{||} \approx (2 \text{ to } 5)E_\perp$ and the diffuse features have $E_{||} \approx 2E_\perp$. These results are consistent with previous measurements of Langmuir wave components, and are in line with theory, where waves in an overdense plasma above the plasma frequency experience Z-infinity resonance at angles with respect to the background magnetic field defined by equation (4). Using this equation with the plasma line and diffuse band frequencies shows that these waves would propagate at angles between 5–20°, which are comparable with the propagation angles produced by the $E_{||}/E_\perp$ ratio values using equation (3).



WHAMP was used to identify the Langmuir/z and whistler/LH surfaces where the plasma line and diffuse feature's wave modes would occur. The $E_{||}/E_\perp$ values are also consistent with the Langmuir/z surface at moderately oblique angles. Wavevector and frequency conservation for a 3-wave process involving the plasma line and diffuse band regions of the dispersion surface are consistent with the third wave being on the whistler/LH surface close to perpendicular propagation and with frequencies close to the LH frequency. The electron and ion data was used to determine instabilities on the LH surface and determined

that the high energy electrons are the more likely source of these waves. The observed electric field energy densities of the whistler/LH waves are large enough, in comparison to the thermal energy density, for a nonlinear process to be viable. The wave energy densities decrease from the whistler/LH waves to the plasma line Langmuir waves to the diffuse band. Comparison of the different wave mode occupation numbers suggest the most plausible explanation is the coalescence of whistler/LH waves $W$ with Langmuir waves $L$ from the plasma line to produce the diffuse band of Langmuir waves $L'$ via the process

$W + L \to L'$. Both the $W$ and $L$ waves are believed to be produced by distinct linear instabilities.

This is similar to the process in Staciewicz et al. [1996], where observation of modulated Langmuir waves suggested these waves were produced through either parametric decay of the primary Langmuir wave into a LH wave and secondary Langmuir waves via the process $L \to L' + W$ or through the scattering of Langmuir waves off pre-existing LH waves via the process $L + W \to L'$, itself obviously a coalescence process. Bonnell at al. [1997] also presented a similar study of modulated Langmuir

waves thought to be produced scattering off electrostatic whistler/LH waves, and showed this was the more likely process than the decay process in their situation. The observations presented here seem to be a similar process to these two studies, of a Langmuir/z mode wave coalescing with or scattering off of the whistler/LH, but here with the Langmuir/z mode waves having significantly weaker amplitudes.

*Data availability.*  https://phi.physics.uiowa.edu/science/tau/data0/rocket/HIBAR_36200/

*Author contributions.*  J.L. was the PI for the HiBAR mission. J.L. and C.M. interpreted the data and experiment. C.M. wrote the codes and performed the data analysis. I.C. contributed the theoretical analysis and all authors contributed to the interpretation of the results. C.M. wrote the manuscript with contributions from all authors.

*Acknowledgements.*  Thanks to the team at Wallops Flight Facility and NASA for supporting the HIBAR payload and launch, as well as engineer Hank Harjes and NASA engineer Bill Payne for instrumentation support. Authors also thank S. Marilia, C. Feltmann, and S.

Bounds for discussions and support. Research at Dartmouth College was supported by NASA grant NNX17AF92G.





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
