# Peer review of "High Bandwidth Measurements of Auroral Langmuir Waves with Multiple Antennas"

_Annales Geophysicae, 2021_

## Author Response (AR2)

Response to Reviewer #1

"High bandwidth measurements of auroral Langmuir waves with multiple antennas" by C. Moser, J. LaBelle, and I. H. Cairns

Comment:

The authors present the analysis via Tables 1 and 2 where the readers may glean that the exact nature of the high-frequency fluctuations is uncertain. It appears that the main focus of Section 3 is on determining whether the measured high-frequency fluctuations are quasi-parallel Langmuir wave or quasi-perpendicular, or oblique, Z/upper-hybrid waves. If the authors can state this physical motivation at the very beginning of Section 3, then it would help readers understand the main thrust of Section 3 a bit more clearly. The authors do mention the motivation of this analysis, but almost at the end of Section 3, line 160, equations 1 & 2. It would be much more helpful for the readers if they mention the purpose of $E_{||}/E_{perp}$ analysis at the beginning (in words) and rephrase in more quantitative details (through equations 1 & 2) at the end of Section 3, and further elaborated in subsequent subsections (as they are laid out herein).

Response:

Thank you for pointing this out. We have added a couple opening sentences to section 3 (lines 148-151) that explains the motivation for the section and why we decided to measure the $E_{||}/E_{perp}$, as well as a sentence at line 185-188 that explains why we calculated the theoretical energy to account for the observations of ratios that are < 1.

We state in the paper that the angles determined by these ratios agree with the theory of Z-mode wave propagation at the Z-infinity resonance angle (lines 217-222).

Comment:

One question that naturally arise is the following: In generating Figure 7 dispersion surfaces the authors have constructed the frequency, $k_{||}$ and $k_{perp}$. This means that the correspond $E_{||}/E_{perp}$ for the Langmuir and diffuse fluctuations can also be computed theoretically. How do they match up with observations, as laid out in the two Tables? are they consistent?

Response:

To generate Figure 7, we assumed $k_{||}/k_{perp} = E_{||}/E_{perp}$ (electrostatic waves). The WHAMP dispersion solver does calculate the theoretical electric field components, and the ratios from that are within 10-25% of the measure values, suggesting that although the waves are not entirely electrostatic, they are close enough for the assumption to be valid. This has been included in the paper at lines 259-264.

Response to Reviewer #2

"High bandwidth measurements of auroral Langmuir waves with multiple antennas" by C. Moser, J. LaBelle, and I. H. Cairns

Comment:

1. The subject of Langmuir turbulence in the auroral ionosphere has been reviewed recently by Akbari, LaBelle and Newman (Front. Astron. Space Sci. 7, 617792, 2021) which should be inserted in the reference. In addition, a discussion should be added to explain the source of field-aligned electron beams that excite the auroral Langmuir waves, and if these substorm events are the ionospheric signatures of magnetic reconnection in the tail regions of the magnetosphere.

Response:

 The Akbari et al. reference was already in the paper, though mis-cited as 2020 rather than 2021. In addition to correcting the citation, we elaborate further on this reference at lines 22-24. We add information about what types of electron distributions are associated with auroral Langmuir waves at lines 36-39. We further elaborate on the geomagnetic conditions at time of launch at lines 75-77.

Comment:

2. In addition to wave-wave processes involving Langmuir and Lower Hybrid waves, the authors should mention other nonlinear wave interaction studies in the auroral ionosphere. For example, Bohm et al. (JGR 95, 12157, 1990) showed that the most intense Langmuir and whistler waves measured by auroral rocket flights occur in association with Alfven waves. The theory of auroral Langmuir-Alfven-whistler events were studied by Chian et al. (A&A 290, L13, 1994) and Lopes and Chian (A&A 305, 669, 1996). In particular, these theoretical papers showed the feasibility of a 3-wave process involving the parametric decay of a Langmuir wave into a whistler wave and an Alfven electromagnetic ion-cyclotron wave which may apply to the observation of electromagnetic ion-cyclotron waves in a flickering aurora by Lund and LaBelle (JGR 17, 241, 1996).

Response:

These are excellent references and show the common occurrences of non-linear processes in the auroral ionosphere. We've supplemented an existing introductory paragraph on nonlinear Langmuir wave phenomena with these references (see lines 46-54).

Comment:

3. Although the approach of this paper is based on the concepts of linear wave dispersion and weakly nonlinear wave-wave interactions, for the sake of completeness a discussion should be inserted to relate these linear and weakly nonlinear wave studies to the description of Langmuir intermittent turbulence consisted of coherent structures such as cavitons discussed by Akbari et al. (JGR Space Phys. 118, 3576, 2013) and phase-space vortices such as electron and ion holes discussed by Ergun et al. (Phys. Rev. Lett. 81, 826, 1998) and Schamel et al. (Phys. Plasmas 27, 062302, 2020).

Response:

The study by Akbari et al. is very interesting and presents observations somewhat similar to those observed in this paper. For completeness, we have added mention of strong turbulence phenomena in the introduction at lines 36-39. We feel it's unlikely that these strong turbulence phenomena relate to our observations, due to the low energy densities of the waves we observe (wave energy density to thermal energy ratio of order 10^-12). For completeness, however, we add a paragraph discussing this topic at lines 317-319.